# Activation of coagulation and proinflammatory pathways in thrombosis with thrombocytopenia syndrome and following COVID-19 vaccination

Malika Aid[1], Kathryn E. Stephenson [1], Ai-ris Y. Collier [1], Joseph P. Nkolola[1], James V. Michael [2], Steven E. McKenzie[2] & Dan H. Barouch [1,3] ✉

Thrombosis with thrombocytopenia syndrome (TTS) is a rare but potentially severe adverse event following immunization with adenovirus vector-based COVID-19 vaccines such as Ad26.COV2.S (Janssen) and ChAdOx1 (AstraZeneca). However, no case of TTS has been reported in over 1.5 million individuals who received a second immunization with Ad26.COV2.S in the United States. Here we utilize transcriptomic and proteomic profiling to compare individuals who receive two doses of Ad26.COV2.S with those vaccinated with BNT162b2 or mRNA-1273. Initial Ad26.COV2.S vaccination induces transient activation of platelet and coagulation and innate immune pathways that resolve by day 7; by contrast, patients with TTS show robust upregulation of these pathways on days 15–19 following initial Ad26.COV2.S vaccination. Meanwhile, a second immunization or a reduced initial dose of Ad26.COV2.S induces lower activation of these pathways than does the full initial dose. Our data suggest a role of coagulation and proinflammatory pathways in TTS pathogenesis, which may help optimize vaccination regimens to reduce TTS risk.

Ad26.COV2.S (Janssen/Johnson & Johnson) is an adenovirus vector-based COVID-19 vaccine[1–3] administered to over 17 million people in the United States, and over 1.5 million people received a second dose of Ad26.COV2.S. Thrombosis with thrombocytopenia syndrome (TTS), also known as vaccine-induced thrombotic thrombocytopenia (VITT), is an extremely rare condition reported in individuals who received Ad26.COV2.S or ChAdOx1[4–6]. Three cases of TTS, including one fatal TTS case, have also been reported following mRNA-1273 immunization[6,7]. Several TTS cases have also been reported following Sinopharm BBIBP-CorV COVID-19 vaccination[8,9].

As of June 12, 2022, 60 TTS cases and 9 deaths were confirmed after initial immunization with Ad26.COV2.S in over 17 million people in the United States, resulting in a TTS rate of 3.53 (95% confidence interval [CI] 2.69–4.53) per million doses and a death rate of 0.53 (95% CI 0.24–1.00) per million doses (Table 1). Following a second immunization with Ad26.COV2.S, no TTS cases (95% CI 0.00–2.41 per million doses) have been reported in over 1.5 million people (Table 1). Moreover, Ad26.COV2.S has been administered at the same dose to over 9.5 million people in South Africa with two mild TTS cases and no TTS deaths reported despite robust pharmacovigilance[10,11]. Anti-vector immunity develops after immunization with Ad26 vectors and is also observed at baseline in the majority of people in South Africa[12], suggesting the possibility that anti-vector immunity may reduce TTS risk.

[1]Center for Virology and Vaccine Research, Beth Israel Deaconess Medical Center, Harvard Medical School, Boston, MA, USA. [2]Department of Medicine, The Cardeza Foundation for Hematologic Research, Thomas Jefferson University, Philadelphia, PA, USA. [3]Ragon Institute of Massachusetts General Hospital, Massachusetts Institute of Technology, and Harvard University, Cambridge, MA, USA. ✉e-mail: dbarouch@bidmc.harvard.edu

To evaluate the host transcriptomic and proteomic changes following vaccination with various COVID-19 vaccines in healthy individuals and in patients who developed TTS, we performed transcriptomic and proteomics profiling in two patients with TTS, twenty healthy individuals who received Ad26.COV2.S, and fourteen healthy individuals who received the mRNA vaccines mRNA-1273 (Moderna) or BNT162b2 (Pfizer) (Fig. 1).

Here we show that Ad26.COV2.S, mRNA-1273, and BNT162b2 vaccination induces transient activation of platelet and coagulation signaling and innate immune pathways, suggesting common host innate immune responses following immunization with these three COVID-19 vaccines. By contrast, in two TTS patients, we observe robust activation of these pathways following vaccination concurrent with clinical illness. A second immunization with Ad26.COV2.S or a reduced initial dose of Ad26.COV2.S results in lower activation of these pathways, which provide insight into TTS pathogenesis and suggest potential strategies to reduce TTS risk following vaccination.

## Results

### Ad26.COV2.S, mRNA-1273, and BNT162b2 induce transient activation of coagulation and innate immune pathways

We assessed transcriptomic profiles following vaccination with Ad26.COV2.S, mRNA-1273, or BNT162b2 by bulk RNA sequencing in peripheral blood (Fig. 1). Samples were evaluated at baseline (D1), one day (D2), and seven days (D8) following initial immunization with $5 \times 10^{10}$ viral particles (vp) or $1 \times 10^{11}$ vp Ad26.COV2.S, and 1 day after prime and boost immunization with mRNA vaccines mRNA-1273 and BNT162b2 (Fig. 2a and Supplementary Fig. 1). Pathway enrichment analysis using a compendium of gene signatures[13–16] (GSEA, $P < 0.05$, $FDR\ q < 0.10$) showed activation of transcriptomic signatures of platelet activation and coagulation (*VCPIP1, TGFB1, SOCS1, SIGLEC1, MARCO, GNB4, FCGR2A, TSPAN2*) as well as innate immune pathways (*BST2, DDXS8, IDO1, IFI16, IFI44, ISG15, MX1*) following Ad26.COV2.S

vaccination that peaked on D2 and was resolved by D8, with lower activation of these pathways at the lower dose of Ad26.COV2.S (Fig. 2a, Wilcoxon signed-rank test $P < 0.05$). Activation of these pathways was similarly observed following vaccination with mRNA-1273 and BNT162b2, particularly after boost immunization and more prominently with mRNA-1273 (Fig. 2a), suggesting that transient activation of these pathways is common for the three vaccines.

To validate these observations at the protein level, we performed serum proteomic profiling using the SomaScan platform[17], which confirmed the activation of these pathways following Ad26.COV2.S, mRNA-1273, and BNT162b2 vaccination (GSEA, $P < 0.05$, $FDR\ q < 0.10$) (Fig. 2b and Supplementary Fig. 2). Consistent with the transcriptomic data, activation of these pathways resolved by D8, with lower activation of these pathways at the lower dose of Ad26.COV2.S (Fig. 2b, Wilcoxon signed-rank test $P < 0.05$). These data show that Ad26.COV2.S, mRNA-1273, and BNT162b2 induced activation of platelet and coagulation pathways and innate proinflammatory signaling one day following vaccination. Activation of these pathways was resolved by 1 week after Ad26.COV2.S vaccination.

### TTS patients show robust and sustained activation of coagulation and innate immune pathways

We next performed proteomic profiling using the SomaScan platform from two patients who developed TTS following Ad26.COV2.S vaccination. Samples were available from the first patient on days 15 and 19 following vaccination and from the second patient on day 16 following vaccination. Cells were not available from these patients for transcriptomic profiling. The first patient was a fatal case of TTS, whereas the second fully recovered. Pathways of platelet activation, coagulation, and innate immune signaling were induced in TTS patients on days 15–19 after Ad26.COV2.S vaccination (Supplementary Fig. 3a). Specifically, in the TTS patients, we observed enrichment of markers of severe COVID-19 ($P = 1.38E\text{-}161$)[18], blood

**Table 1 | TTS cases and deaths after prime and boost immunization with Ad26.COV2.S as of June 12, 2022 in the United States**

| | Number of doses (million) | TTS Events | Number of TTS events per million doses (95% CI) | Number of deaths | Number of deaths per million doses (95% CI) |
|---|---|---|---|---|---|
| Prime | 17,015,922 | 60 | 3.53 (2.69, 4.53) | 9 | 0.53 (0.24, 1) |
| Boost | 1,528,813 | 0 | 0 (0, 2.41) | 0 | 0 (0, 2.41) |

*CI* confidence interval.
TTS (and death) rate per million doses was estimated using a Poisson distribution.

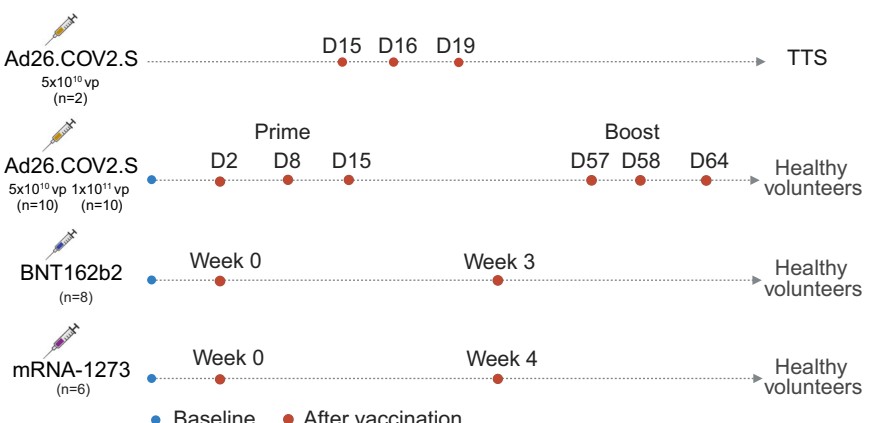

**Fig. 1 | Study design.** Peripheral blood was collected from healthy individuals prior to vaccination and following prime and boost immunization with Ad26.COV2.S (*n* = 20), BNT162b2 (*n* = 8), and mRNA-1273 (*n* = 5) for transcriptomic profiling using bulk RNA sequencing. Samples were collected from two TTS patients on days 15, 16, and 19 following Ad26.COV2.S vaccination and from healthy individuals following prime and boost immunization with Ad26.COV2.S (*n* = 20), BNT162b2 (*n* = 5), and mRNA-1273 (6) for proteomics profiling using the SomaScan platform. Created with BioRender.com.

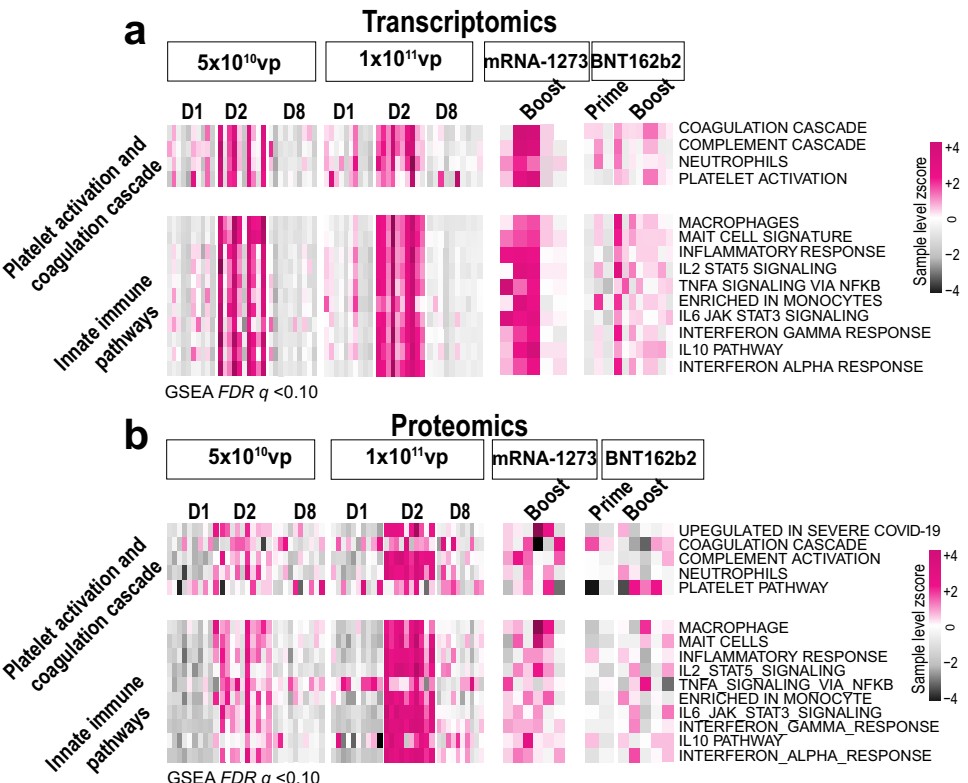

**Fig. 2 | Transcriptomics and proteomics profiling following Ad26.COV2.S, BNT162b2, and mRNA-1273 vaccination in healthy individuals.**
**a, b** Transcriptomics (**a**) and proteomics (**b**) signatures of platelet activation and coagulation cascades and innate immune pathways significantly increased (GSEA: FDR q value < 0.10) on day one (D2) following Ad26.COV2.S 5 × 10^10 vp (n = 10 or

Ad26.COV2.S 1 × 10^11 vp (n = 10) and after boost immunization with mRNA-1273 and BNT162b2 compared. Heatmaps show the pathway SLEA Z-score, where each row corresponds to a pathway and each column represents an individual sample−color gradient ranging from dark (downregulated) to dark pink (upregulated) genes and proteins. Source data are provided as a Source data file.

coagulation (P = 5.62E-23), activation of complement (P = 1.13E-21), platelet (P = 1.05E-09), neutrophil (P = 2.07E-07), MAIT cell activation (P = 0.00008), macrophage (P = 0.0001), inflammatory response (P = 0.0008), TNF signaling via NF-κB (P = 0.0009), and IL6 JAK STAT3 signaling (P = 0.001). Markers of proinflammatory signaling (*IL18, IL1RL1, IRF1, IFI16, CXCL13, IL21R*), neutrophil activation (*NCF1, NCF2*), and coagulation cascade (*VWF, SVEP1, THBS1, F9, THSD7A, ADAMTS6, FGFRL1*) were higher in the TTS patients compared with Ad26.COV2.S vaccine recipients' healthy controls (Supplementary Fig. 2 and Supplementary Data 1).

To further explore pathways associated with TTS pathogenesis, we compared the proteomic profile in the TTS patients on days 15–19 and the healthy Ad26.COV2.S vaccine recipients on day 15 following immunization. Unsupervised analysis showed a cluster of Ad26.COV2.S-vaccinated healthy individuals and a cluster of the TTS cases (Supplementary Fig. 3b). On day 15 following Ad26.COV2.S vaccination, we observed resolution of cytokines and proinflammatory signaling, platelet activation and coagulation cascade, and markers associated with COVID-19 severity[18], whereas these signatures were markedly elevated in TTS patients (Supplementary Fig. 3c–e). The TTS patients showed significant upregulation of proinflammatory signaling (*IRF1-6, IL1B, IL1O, IL21R, IL18, CSF, IL17D, IL1RL1, STAT1, CAPG, MIF*), apoptosis (*BCL2, FOXO3*), and platelet activation and coagulation (*VWF, GP6, MPO, PF4, PCAM1, PDGFD, THBS1, THBS2*). Upregulation of these markers was also observed in a patient with TTS after vaccination with ChAdOx1[19], and recent work has suggested that TTS may be related to persistent proinflammatory signaling, platelet activation, and neutrophil extracellular trap (NET) formation[20,21]. In contrast, we observed downregulation of transforming growth factor beta (*TGFBR3*), natural killer markers (*KLRC3, KIR2DS2, KLRF1*, and neural

cell adhesion molecules *NCAM1* and *NCAM2* in the TTS patients (Supplementary Fig. 3f). The analysis of the upstream regulators of the top proteins increased in TTS showed the enrichment of markers associated with transcriptional and immune regulation, apoptosis, complement, angiogenesis, and coagulation, likely contributing to TTS pathogenesis (Supplementary Fig. 4).

**Reduced activation of coagulation and innate immune pathways following Ad26.COV2.S vaccination in the presence of anti-vector immunity**

We next compared the transcriptomic and proteomic changes induced by Ad26.COV2.S one day following the second immunization (D58) compared with one day following the initial immunization (D2) (Fig. 3). As described above, no cases of TTS have been reported in over 1.5 million people who received a second immunization of Ad26.COV2.S (Table 1). Activation of coagulation, complement, platelet, macrophage, proinflammatory, cytokine, and interferon pathways was consistently lower (GSEA FDR q < 0.10) on D58 compared with D2 by both transcriptomics (Fig. 3a, b and Supplementary Data 2) and proteomics (Fig. 3c, d and Supplementary Data 2). These differences were greater with the higher dose of Ad26.COV2.S. In particular, interferon-stimulated genes (*MX1, IRF7, ISG15, OAS1, OAS2*) and proinflammatory markers, cytokines, and chemokines (*IL1RN, CXCL10, IL6, CRP, IFN-γ*) were significantly lower (Wilcoxon signed-rank test P < 0.05) following the boost compared with the prime immunization by both transcriptomics in peripheral blood (Fig. 4a) and proteomics in serum (Fig. 4b, c). These data show that a second immunization with Ad26.COV2.S in the setting of anti-vector immunity resulted in a reduced inflammatory response, which is consistent with the lower reactogenicity observed following a second dose of Ad26.COV2.S[1].

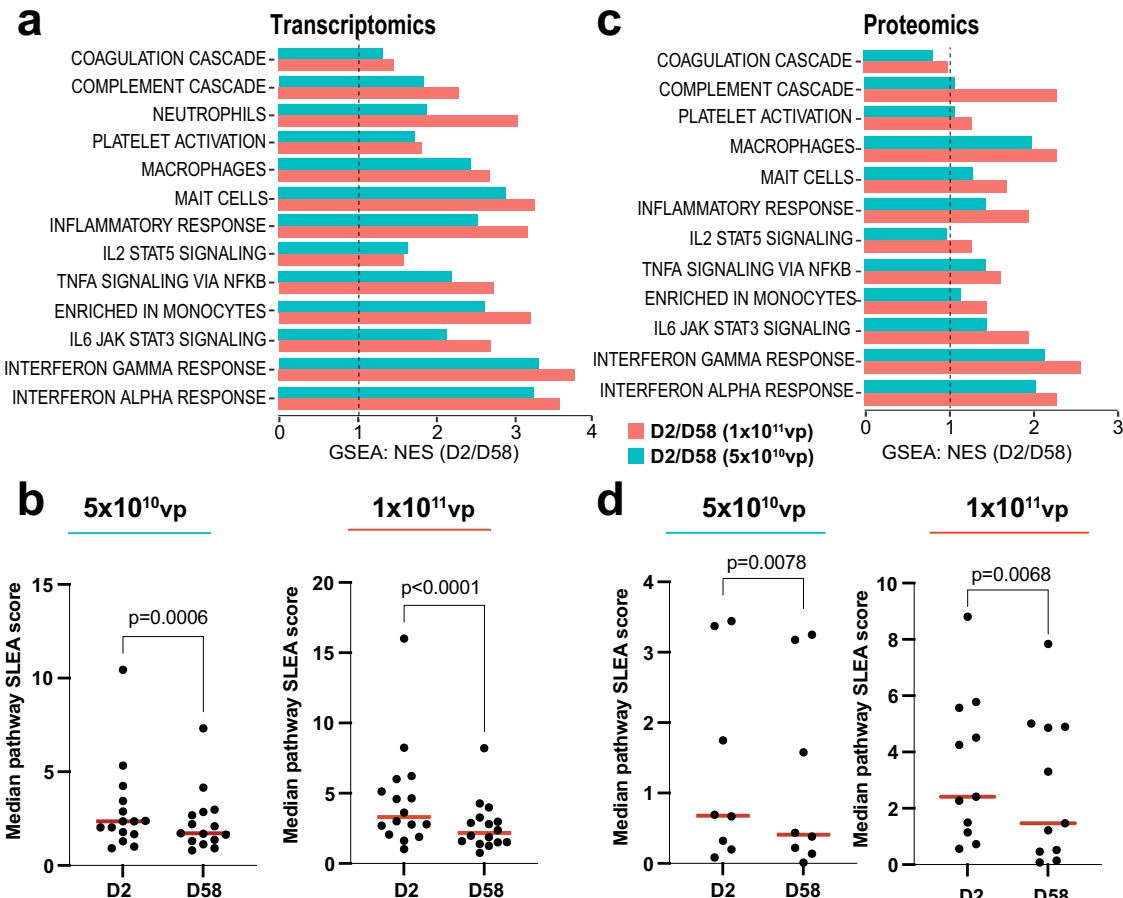

**Fig. 3 | Reduced activation of innate immune pathways following Ad26.COV2.S vaccination in the presence of anti-vector immunity. a, c** GSEA normalized enrichment score (NES) ratio (D2/D58) of innate immune pathways and coagulation cascades following initial immunization (D2) compared with boost immunization (D58) with Ad26.COV2.S $5 \times 10^{10}$ vp (cyan) or $1 \times 10^{11}$ vp (orange) in the blood (**a**) or in serum (**c**). **b, d** Scatter plots of median sample-level enrichment analysis (SLEA) score of pathways in (**a**, **c**). Individual dots represent the pathway median SLEA score across all vaccinated participants. Comparison of pathways between D2 and D58 were evaluated using a matched two-tailed Wilcoxon signed-rank test and a $P$ value cutoff of <0.05. Exact $P$ values were shown for each comparison. Red bars correspond to the median of all scores for each time point. Source data are provided as a Source data file.

To generalize these observations using a different Ad26 vector-based vaccine, we assessed transcriptomic and proteomic profiles in 12 healthy individuals who were enrolled specifically with or without Ad26 vector immunity and were then vaccinated with a single dose of $5 \times 10^{10}$ vp Ad26.HIV.EnvA[22]. Seven participants were Ad26 seronegative at baseline, whereas five were Ad26 seropositive at baseline (Supplementary Fig. 5a). Upregulation of innate immune pathways following Ad26.HIV.EnvA vaccination was higher in the baseline Ad26 seronegative individuals than in the baseline Ad26 seropositive individuals by transcriptomics (Fig. 5a) and proteomics (Fig. 5b, c). Similar to Ad26.COV2.S vaccination (Fig. 2), pathways of platelet activation and coagulation, and innate immune pathways were significantly upregulated (GSEA *FDR q* < 0.10) on D2 and resolved by D8 following Ad26.HIV.EnvA vaccination by transcriptomics (Supplementary Fig. 5b) and proteomics (Supplementary Fig. 5c).

Next, we compared the transcriptomic and proteomic profiles of $5 \times 10^{10}$ vp Ad26.COV2.S with $5 \times 10^{10}$ vp Ad26.HIV.EnvA (Supplementary Fig. 6) and observed higher levels of transcriptomic and proteomic (GSEA *FDR q* < 0.10) signatures of platelet, complement, coagulation, and innate immune pathways with Ad26.COV2.S at the mRNA (Fig. 5d) and the protein (Fig. 5e) levels. These observations suggest that the Ad26 vector and the SARS-CoV-2 Spike protein likely contribute to the innate inflammatory responses after Ad26.COV2.S vaccination.

**Reduced activation of coagulation and innate proinflammatory pathways with lower doses of Ad26.COV2.S in rhesus macaques**

We previously reported boosting with $1 \times 10^{10}$, $2.5 \times 10^{10}$, or $5 \times 10^{10}$ vp Ad26.COV2.S in humans resulted in comparable immune responses, suggesting the feasibility of using lower Ad26.COV2.S doses for boosting[23]. We showed reduced activation of the coagulation and innate proinflammatory pathways with $5 \times 10^{10}$ vp compared with $1 \times 10^{11}$ vp Ad26.COV2.S in humans (Figs. 2, 3, and 5), suggesting a dose effect. To explore whether lower doses of Ad26.COV2.S would further reduce activation of the coagulation and innate proinflammatory pathways; we performed transcriptomic and proteomic profiling in 25 rhesus macaques (5 per group) vaccinated with sham saline or $2 \times 10^9$, $1.125 \times 10^{10}$, $5 \times 10^{10}$, or $1 \times 10^{11}$ vp Ad26.COV2.S[24]. We observed a dose-dependent upregulation of platelet and coagulation and innate immune pathways on D2 with complete resolution by D8 (Supplementary Fig. 7a, b). In particular, we observed decreased activation of these pathways with the $1.125 \times 10^{10}$ vp dose compared with the $5 \times 10^{10}$ vp and $1 \times 10^{11}$ vp doses (Supplementary Fig. 7c–e). These data show that vaccination with a lower dose of Ad26.COV2.S (GSEA *FDR q* < 0.05) led to reduced activation of coagulation and innate immune pathways.

Taken together, our data suggest that individuals with anti-vector immunity that received Ad26.COV2.S effectively received a lower effective dose of Ad26.COV2.S, which may have contributed to the lower TTS risk (Table 1). It is also possible that anti-vector NAbs may

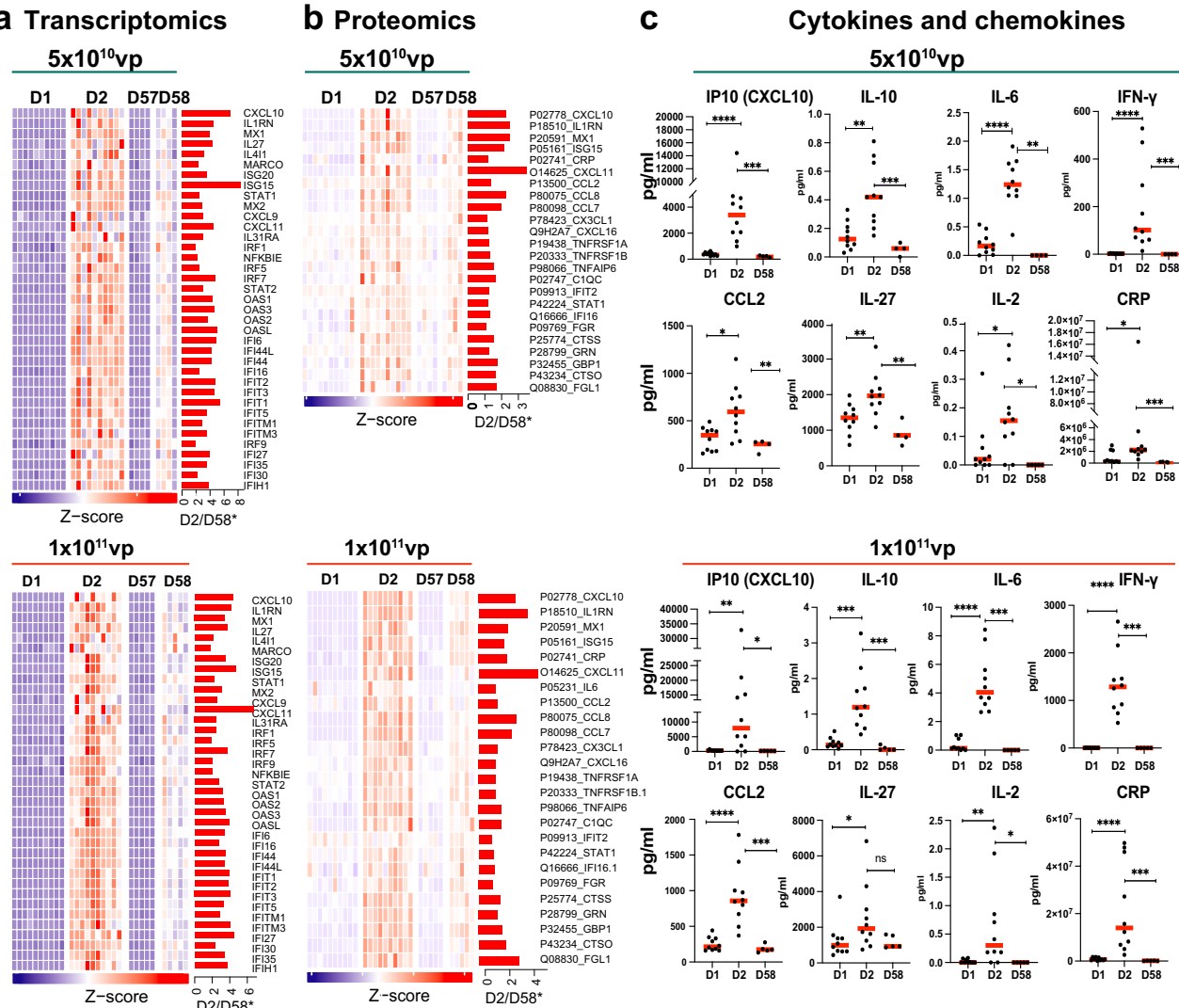

**Fig. 4 | Reduced activation of innate immune pathways following Ad26.COV2.S vaccination in the presence of anti-vector immunity. a–c** Transcriptomics (**a**) and serum proteomics measured by SomaO (**b**) or mesoscale platform (**c**), markers on D1 (baseline) and following initial immunization (D2) compared with boost immunization (D58) for low and high dose of Ad26.COV2.S. **a** heatmaps show the row-scaled expression in blood (Wald test, adjusted P < 0.05) or serum levels (limma t test, adjusted P < 0.05) of the top interferon and proinflammatory markers increased on D2 compared with D58. The barplots shown on the right side of each heatmap correspond to the log2 fold change mRNA ratio of D2/D58 for each marker. Comparison of individual marker serum levels was evaluated using a matched two-tailed Wilcoxon signed-rank test and a P value cutoff of <0.05, where *<0.05; **<0.005, and ***<0.0005. Exact P values were shown for each comparison. Red bars correspond to the median of all scores for each time point. Source data are provided as a Source data file.

reduce Ad26 binding to platelet factor 4 (PF4)[25] (Supplementary Fig. 8). Moreover, pathways of innate immune cell signatures, metabolism, and transcription regulation (Supplementary Fig. 9a, b) were similarly upregulated in both humans and macaques following immunization with Ad26.COV2.S or Ad26.HIV.EnvA, suggesting the utility of the preclinical model in nonhuman primates for analyzing inflammatory and innate responses following vaccination.

## Discussion

Our knowledge of the etiology and pathogenesis of TTS remains incomplete. In this study, we show that Ad26.COV2.S vaccination induced transient activation of platelet and coagulation signaling and innate immune pathways, which peaked one day following vaccination and resolved by one week following vaccination. Similar activation of these pathways was observed after mRNA-1273 and BNT162b2 boost immunization, suggesting common host innate immune responses following these three COVID-19 vaccines. In contrast, in two TTS patients, we observed activation of these pathways on days 15–19

following vaccination, concurrent with clinical illness. A limitation of our study is that TTS is extremely rare, and we could only study 2 TTS patients. Thus, our findings require confirmation in other cohorts for generalizability.

It is currently unknown which factors trigger the cascade of events that lead to TTS in rare individuals who receive Ad26.COV2.S or ChAdOx1, as well as in occasional individuals who receive mRNA and inactivated virus-based COVID-19 vaccines. We speculate that activation of platelet and coagulation signaling and innate immune pathways is a necessary but not sufficient condition for TTS, since we did not observe greater induction of these pathways by Ad26.COV2.S compared with mRNA-1273 or BNT162b2. The additional factors that trigger TTS in rare individuals remain to be defined. The potential contribution of Spike-specific immune responses to the development of TTS also remains to be determined.

In contrast with the rare cases of TTS following initial Ad26.COV2.S vaccination, it is important to note that no case of TTS has been reported following over 1.5 million second immunizations

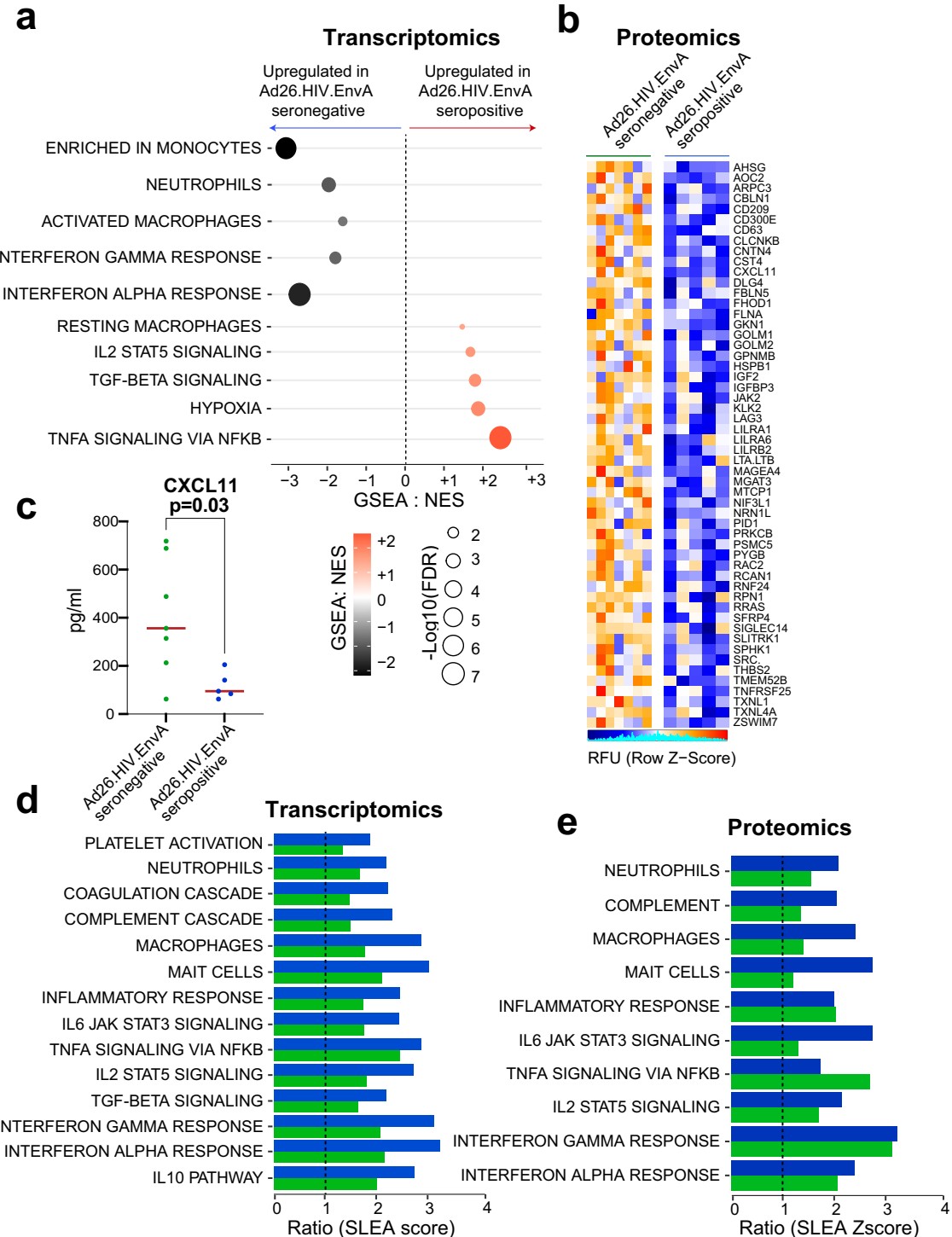

**Fig. 5 | Reduced activation of innate immune pathways following Ad26.HIV.EnvA vaccination in the presence of anti-vector immunity. a** Innate immune pathways in peripheral blood in the Ad26.HIV seropositive (*n* = 5) compared with Ad26.HIV seronegative (*n* = 7) groups (GSEA, FDR *q* value < 0.10). The circle color represents pathway enrichment scores (NES) ranging from blue (decreased) to red (increased), and circle sizes represent the −log10 (FDR *q* value). **b, c** Serum levels of innate proinflammatory markers in the Ad26.HIV.EnvA seropositive compared with Ad26.HIV seronegative group by proteomics. **b** shows the SomaO serum level of significantly upregulated proteins (Limma *t* test, adjusted BH-*P* < 0.05) in the seronegative compared with the seropositive group. **c** shows the serum level of *CXCL11* measured by the mesoscale assay, where comparison between groups was assessed using the two-tailed Wilcoxon signed-rank test; exact *P* value is shown. **d, e** Innate immune pathways and coagulation cascade in peripheral blood (**d**) and serum (**e**) 1 day (D2) following immunization with Ad26.HIV.EnvA seronegative (in green) or Ad26.HIV.EnvA seropositive (in blue) vaccinated individuals. Barplots show the ratio Ad26.COV2.S/Ad26.HIV.EnvA of pathways SLEA score for seronegative (in green) and seropositive (in blue). Source data are provided as a Source data file.

with Ad26.COV2.S in the United States (Table 1), suggesting that these individuals have factors that may be protective against TTS. Moreover, there have been only two reported cases of TTS and no TTS deaths in over 9.5 million individuals who received initial immunization with Ad26.COV2.S in South Africa, which has excellent pharmacovigilance, as well as no reported TTS deaths in over 90 million individuals who received initial immunization with Ad26.COV2.S in other countries in Africa, where most individuals have low levels of Ad26 vector immunity[12]. Moreover, there is a markedly reduced rate of TTS in people who received a second immunization with ChAdOx1 compared with the first immunization with ChAdOx1[26], as well as reduced TTS with ChAdOx1 in the developing world[27] where there are high levels of baseline anti-adenovirus immunity. These observations suggest that anti-vector immunity may reduce TTS risk for both Ad26.COV2.S and ChAdOx1, possibly by reducing the effective vaccine dose and decreasing the subsequent inflammatory response, or alternatively by interfering with vector binding to PF4[25].

Our data show reduced activation of platelet and coagulation signaling and innate immune pathways in people receiving a second immunization with Ad26.COV2.S, in people with baseline anti-vector immunity receiving an initial immunization with Ad26.HIV.EnvA, and in humans and macaques without baseline anti-vector immunity receiving a reduced dose of Ad26.COV2.S. Anti-vector immunity following initial Ad26.COV2.S immunization results in a reduction of the effective vaccine dose, as suggested by the reduced reactogenicity following a second Ad26.COV2.S immunization[1]. These data suggest that a lower dose of Ad26.COV2.S may reduce TTS risk, which can be done without compromising immunogenicity[23]. Regardless, our data provide insight into the pathogenesis of TTS and the activation of inflammatory pathways in healthy individuals and TTS patients following vaccination.

## Methods
### Clinical cohorts
The Ad26.COV2.S clinical trial was conducted at a single site at Beth Israel Deaconess Medical Center in Boston and has been published[2]. The protocol was approved by the Beth Israel Deaconess Medical Center Institutional Review Board. All participants gave written informed consent and successfully completed an assessment of understanding before the initiation of study procedures. Eligible participants were 18–55 years old (inclusive) and negative for SARS-CoV-2 infection by screening nasopharyngeal polymerase chain reaction and serum immunoglobulin testing. Eligible participants also had a body mass index of 30 or less (calculated as weight in kilograms divided by height in meters squared); were healthy, in the investigator's clinical judgment, as confirmed by medical history, physical examination, clinical laboratory assessments, and vital signs performed at screening; and did not have comorbidities related to an increased risk of severe COVID-19. Twenty-five participants were randomized to receive 1 or 2 intramuscular injections with $5 \times 10^{10}$ viral particles or $1 \times 10^{11}$ viral particles of Ad26.COV2.S vaccine or placebo administered on day 1 and day 57 (5 participants in each group). Peripheral blood, serum, and plasma were collected at baseline before vaccination (D1), and on days 1 and 7 following the first dose. Plasma samples were additionally collected on day 15 (D15) following first dose in all vaccinated individuals. Half of the participants were boosted on day 57, and peripheral blood and serum samples were collected on days 58 and 64 following boost immunization.

We enrolled individuals 18 years old and older in a specimen biorepository at Beth Israel Deaconess Medical Center. The protocol was approved by the Beth Israel Deaconess Medical Center Institutional Review Board. This cohort included individuals vaccinated with BNT162b2 or mRNA-1273 and provided samples following their first and/or second vaccine doses. All participants provided informed consent. Individuals were excluded if they received immunosuppressive

medications or had a history of SARS-CoV-2 infection. PBMC and serum were collected from all participants at prime and boost immunization.

The Ad26.HIV.EnvA clinical trial was conducted at a single site at Brigham and Women's Hospital in Boston and has been published[22]. This clinical trial involved a single intramuscular injection of Ad26-EnvA at a dose of $5 \times 10^{10}$ viral particles in healthy adults. Twenty-four subjects were enrolled: 16 Ad26 seronegative and 8 Ad26 seropositive. Subjects were healthy, HIV-uninfected, between 18 and 50 years of age, able to complete an assessment of understanding questionnaire for participation, and were at low risk for HIV acquisition. This study was approved by the Partners Institutional Review Board and Institutional Biosafety Committee, and written informed consent was obtained from all subjects. PBMC and serum samples were collected from 7 Ad26 seronegative and 5 Ad26 seropositive individuals at baseline (D1), and on days 1 (D2) and 7 (D8) following vaccination.

### Animal cohorts
In total, 30 outbred Indian-origin adult 5–8 kg male (10) and female (20) rhesus macaques (*Macaca mulatta*) were randomly allocated to groups[24]. All animals were housed at Bioqual, Inc. (Rockville, MD) and were caged individually or paired. Animals received a single immunization of $1 \times 10^{11}$, $5 \times 10^{10}$, $1.125 \times 10^{10}$, or $2 \times 10^{9}$ viral particles (vp) of Ad26.COV2.S (n = 5/group) or sham (n = 10) by the intramuscular route without adjuvant at week 0. Animal studies were conducted in compliance with all relevant local, state, and federal regulations. All animal studies were approved by the Institutional Animal Care and Use Committee (IACUC) at Bioqual. Blood, PBMC, and serum samples were collected at baseline (D1) and on days 1 (D2), 7 (D8), and 14 (D15) following vaccination.

### Samples processing and bulk RNA sequencing
Total RNA was extracted from whole blood samples using the PAXgene Blood RNA Kit IVD (PreAnalytix). RNA quality was assessed with the Fragment Analyzer (Agilent) and its Standard Sensitivity RNA kit. Total RNA was normalized to 100 ng before random hexamer priming and libraries were generated by the TruSeq Stranded Total RNA–Globin kit (Illumina). The resulting libraries were assessed on the Fragment Analyzer (Agilent) with the High Sense Large Fragment kit and quantified using a Qubit 3.0 fluorometer (Life Technologies). Medium-depth sequencing, with at least 30 million reads per sample, was performed on a NovaSeq 6000 (Illumina) with a PE100 configuration, paired-end run yielding ~40 million reads per sample. Peripheral blood mononuclear cell (PBMC) were collected from the Ad26.HIV.EnvA vaccinated individuals. PAXgene Blood samples were not available for the Ad26.HIV.EnvA vaccinated individuals. To compare the reproducibility of PAXgene Blood and PBMC transcriptomic, PBMC were collected from the same cohort of rhesus macaques for bulk RNA sequencing.

Cryopreserved PBMC samples were thawed, counted, and assessed for viability, then 1 million PBMCs were pelleted and lysed in 350 uL of RLT with beta-mercaptoethanol. RNA was isolated using the RNeasy Mini kit (Qiagen). RNA quality was assessed with the Fragment Analyzer (Agilent) and its Standard Sensitivity RNA kit. Total RNA was used to prepare libraries using the TruSeq Stranded Total RNA Globin kit (Illumina). The resulting libraries were assessed on the Fragment Analyzer (Agilent) with the High Sense Large DNA Fragment kit and quantified using a Qubit 3.0 fluorometer (Life Technologies). Medium to high-depth sequencing (>40 million reads per sample) was performed on the NovaSeq 6000 platform (Illumina) with a 100-base pair, paired-end run design.

The quality of the raw reads was assessed using multiQC[28]. Raw demultiplexed fastq paired-end read files were trimmed of adapters and filtered using the program skewer to discard those with an average phred quality score of less than 30 or a length of less than 36. Human-sequenced data were aligned to the Homo Sapiens NCBI reference

genome assembly version GRCh38, and the rhesus macaques data were aligned to the Mmul10-100 and the MacaM assemblies and annotations of the Indian rhesus macaque genome using STAR version 2.7.3a[29]. Transcript abundance estimates were calculated internal to the STAR aligner using the algorithm of htseq-count. DESeq2[30] was used for normalization, producing both a raw and normalized read count table. DESeq2 was also used to generate the differentially expressed genes for all bulk RNA-Seq data sets. Differentially expressed genes were assessed using Wald test with a significance cutoff of $P < 0.05$. All DEGs with mean of normalized counts for all samples (basemean) <10 were filtered. All $P$ values are corrected for multiple testing using the BH method cutoff of <0.05.

## SomaScan proteomics

In all, 55 µl serum or plasma from participants, five pooled serum controls, and one buffer control were analyzed using the SomaScan Assay Kit for human serum V4.1 (Cat#. 900-00021), measuring the expression of 6596 unique human protein targets using 7596 SOMA-mer (slow off-rate modified aptamer) reagents, single-stranded DNA aptamers, according to the manufacturer's standard protocol (Soma-Logic; Boulder, CO). The modified aptamer binding reagents, SomaS-can Assay, its performance characteristics, and specificity to human targets have been previously described[17]. The assay used standard controls, including 12 hybridization normalization control sequences used to control for variability in the Agilent microarray readout process, as well as five human calibrator control pooled serum replicates and 3 Quality Control (QC) pooled replicates used to mitigate batch effects and verify the quality of the assay run using standard acceptance criteria. The readout is performed using Agilent microarray hybridization, scan, and feature extraction technology.

Twelve Hybridization Control SOMAmers are added alongside SOMAmers to be measured from the serum samples and controls of each well during the SOMAmer elution step to control for readout variability. The control samples are run repeatedly during assay qualification and robust point estimates are generated and stored as references for each SOMAmer result for the Calibrator and QC samples. The results are used as references for the SomaScan V4.1 (or SomaScan V4.0) Assay. Plate calibration is performed by calculating the ratio of the Calibrator Reference relative fluorescence unit (RFU) value to the plate-specific Calibrator replicate median RFU value for each SOMAmer. The resulting ratio distribution is decomposed into a Plate Scale factor defined by the median of the distribution and a vector of SOMAmer-specific Calibration Scale Factors. Normalization of QC replicates and samples is performed using Adaptive Normalization by Maximum Likelihood (ANML) with point and variance estimates from a normal U.S. population. Post-calibration accuracy is estimated using the ratio of the QC reference RFU value to the plate-specific QC replicate median RFU value for each SOMAmer. The resulting QC ratio distribution provides a robust estimate of accuracy for each SOMAmer on every plate. Plate-specific Acceptance Criteria are as follows: Plate Scale Factor between 0.4 and 2.5 and 85% of QC ratios between 0.8 and 1.2. For platform standardization, all RFU values across all samples were converted to the SomaScan V4.0 using Somascan internal scaling factors. For SomaO reference serum and plasma data sets, EDTA plasma and serum baseline samples were collected from individuals without any underlying conditions and represent a healthy adult proteome from a mix of Male and Female individuals ranging from 18–90 years of age. SomaScan RFU values and clinical information are obfuscated to protect personally identifiable information while preserving biologically relevant biomarkers. These reference data sets were provided by SomaLogic.

We used the Linear Models for Microarray Data (Limma) R package[31] to identify differentially expressed proteins. The method involves fitting a linear model to the data and then performing a $t$ test to identify proteins that are differentially expressed between two or more groups. P values were corrected for multiple testing using the Benjamini−Hochberg method. Given the small number of TTS patients, to identify significantly increased or decreased proteins in the TTS patients, we used the following method: for each time point, we first normalized each RFU value using the Z-score. Proteins were then ranked by their Z-score from high to low. The Z-score value indicates how far the protein level is from the median expression of all 7000 SOMAmers. High Z-scores correspond to proteins with high RFU levels. We selected the top 90th percentile for each time point as the top highly expressed proteins. To obtain a baseline level of each protein (SoMAmer), we averaged the serum level of each protein collected from individuals without any underlying conditions from a healthy adult population. Next, for each protein, we measured the ratio TTS/baseline. We selected all proteins with a ratio (TTS/baseline) of >2.

## Serum and plasma cytokines and chemokines profiling

ELISA (Mesoscale) Serum and plasma cytokines and chemokines profiling was performed using the commercial V-PLEX Plus Human Biomarker 46-Plex Kit, SECTOR (5 PL) assay (Meso Scale MULTI-ARRAY Technology) available by Meso Scale Discovery (MSD).

The assay was performed according to the manufacturer's instructions (https://www.mesoscale.com/en/technical_resources/technical_literature/technical_notes_search). The V-PLEX Human Biomarker 46-Plex commercial Kit (https://www.mesoscale.com/en/products/v-plex-plus-human-biomarker-46-plex-kit-k15088g/) includes 46 biomarkers in inflammation, chemotaxis, angiogenesis, and immune system regulation. In total, 300 µL of plasma or serum from each donor was combined with the biotinylated antibody plus the assigned linker and the SULFO-TAG conjugated detection antibody; in parallel, a multi-analyte calibrator standard was prepared by doing fourfold serial dilutions. Both samples and calibrators were mixed with the read buffer and loaded in a 10-spot V-PLEX plate, which was read by the MESOQuickPlex SQ 120. Samples were measured in duplicates. The plasma and serum cytokines and chemokines values (pg/mL) were extrapolated from the standard curve of each specific analyte. All values are given in pg/mL based on the calibration standard curve. Undetectable analytes out of the calculation range of the standard curve determined by the MSD software. For downstream analysis, all #values were replaced by the minimal value in the column for that analyte. GraphPad Version 9.4.0 (453), and a paired non-parametric Wilcoxon test were used to assess the significant increase or decrease of each analyte after vaccination compared to baseline. Scatter plots of increased analytes were performed using GraphPad Version 9.4.0 (453).

## Surface plasmon resonance

Binding assays were performed using a Biacore 3000 (Cytiva) and HBS-EP+ immobilization buffer [0.1 M HEPES, 1.5 M NaCl, 0.03 M EDTA and 0.5% v/v Surfactant P20 (Cytiva)]. Ad26 at -3 × 10^11 viral particles was diluted 1:5 in acetate 4.5 buffer (Cytiva) and immobilized to -500 response units on a C1 sensor chip using a standard amine coupling protocol. Polyclonal pre-Ad26 or post-Ad26 vaccination human serum diluted 1:5 in HBS-EP+ immobilization buffer was flowed over bound Ad26 at 30 µL/min for 1 min with a 2 min stabilization time. Threefold dilutions (3000, 1000, 333, 111, and 37 nM) of recombinant human platelet factor 4 (PF4) (Abcam) were titrated over the sensor chip with an association time of 60 s and a dissociation time of 60 s at a flow rate of 50 µL/min. The surface was regenerated with a 30-s injection of 25 mM NaOH at a flow rate of 50 µL/min. All sensorgram plots were subtracted from the reference flow cell and buffer cycle to remove non-specific responses. The KD values were calculated assuming a 1:1 interaction using BIAevluation software.

## Bioinformatics analysis and statistical methods

Analyses of differentially expressed proteins were performed using the limma R package[31]. DESeq2 R package was used to generate the

 

differentially expressed genes for all bulk RNA-Seq data sets. Significantly upregulated or downregulated proteins and genes were selected using a Benjamini–Hochberg adjusted *P* value of <0.05. Gene Set Enrichment Analysis (GSEA) throughout the study was performed to assess enrichment in pathways of transcriptomic and proteomics significant markers modulated following vaccination with Ad26.COV2.S and mRNA vaccines. Briefly, genes were pre-ranked, using the limma t statistic (proteomics) and the DESeq2 stats (transcriptomic), and the significant enrichment of various genesets from MSigDB[13] and the BTM modules database[15] were tested using GSEA pre-ranked list with the following parameters: number of permutation =1000; Max size=5000; min Size=10; Enrichment statistic =weighted; Collapsing mode=Abs_max_of_probe; Normalization mode= meandiv).

We used a false discovery rate FDR *q*-value threshold of 0.10 to select significant pathways enriched in genes or proteins increased or decreased in the different comparisons. Leading edge genes of significant pathways were selected for Sample-level enrichment analysis SLEA analysis. SLEA[32] was used to quantify the enrichment of each pathway within each sample. Briefly, the expression of all the genes in a specific pathway is averaged across each individual and compared to the average expression of 1000 randomly generated genesets of the same size. The resulting averaged expression was then scaled using Z-scores to reflect the overall perturbation of each pathway in each sample. All p values were adjusted for multiple testing using the Benjamini–Hochberg method and a cutoff of 0.05. Pathway enrichment analysis of top significant proteins in TTS patients selected using the 90th percentile method was performed using the GeneOverlap R package to test the enrichment of these proteins in pathways of interferon, inflammatory signaling, platelet, and coagulation cascade, and proteins increased in COVID-19 severe patients. The upstream regulation analysis shown in Extended Data Fig. 4 was performed using EnrichR[33]. Statistical evaluation of pathways enrichment was performed using GraphPad Prism Software version 9.4.0 and the non-parametric Kruskal-Wallis test by comparing the mean rank across all samples. Where applicable, *P* values were corrected for multiple testing using Dunn's test. Comparison of pathways and cytokines between Ad26.COV2.S prime (D2) and boost (D58) were evaluated using Wilcoxon signed-rank test. All figures were generated using Integrated Development for R, RStudio 2022.02.0 Build 443.

### Reporting summary

Further information on research design is available in the Nature Portfolio Reporting Summary linked to this article.

## Data availability

The data discussed in this publication have been deposited in NCBI's Gene Expression Omnibus and are accessible through GEO Series accession number GSE220659. All data are included in the Supplementary Information or available from the authors upon reasonable requests, as are unique reagents used in this Article. The raw numbers for charts and graphs are available in the Source Data file whenever possible. Source data are provided with this paper.

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

## Acknowledgements
We thank Monica Ianosi-Irimie, Samuel Vidal, Cindy Wu, and Towia Libermann for generous reagents, assistance, and advice. We acknowledge funding from NIH grant CA260476, BARDA contract HHSO100201700018C, Massachusetts Consortium for Pathogen Readiness, Ragon Institute, and Janssen.

## Author contributions
D.H.B. and M.A. designed the study and performed the analysis. K.E.S., J.P.N. and A.Y.C. enrolled the clinical cohorts. S.McK. and J.V.M. provided TTS samples and input on the study design. D.H.B. and M.A. wrote the paper with all co-authors.

## Competing interests
D.H.B. is a co-inventor on provisional vaccine patents licensed to Janssen (63/121,482; 63/133,969; 63/135,182). S.McK. is on the Scientific Advisory Board of Veralox Therapeutics. The remaining authors declare no competing interests.
