## [Peer Review File · Nature Communications]

Activation of Coagulation and Proinflammatory Pathways in Thrombosis with Thrombocytopenia Syndrome and Following COVID-19 VaccinationEditorial Note: This manuscript has been previously reviewed at another journal that is not operating a transparent peer review scheme. This document only contains reviewer comments and rebuttal letters for versions considered at *Nature Communications*.

REVIEWERS' COMMENTS

Reviewer #1 (Remarks to the Author):

The revised version of the manuscript has improved. The data on the robust stimulation of platelet activation and coagulation pathways and innate immune pathways is convincing. Also the data showing reduction in activation of these pathways during second vaccination and in case of a reduced vaccine dose are clear.

The additional experiments showing inhibition of PF4 binding to the vaccine vector in the presence of sera of vaccinated individuals contributes to better understanding the mechanisms underlying the different immune response patterns towards PF4. However, while reduced binding of PF4 to which the vaccine vector may explain reduced formation of anti-PF4 antibodies.

The weak in part of the manuscript are in the studies on TTS in patients. The manuscript still states that "TTS patients showed a robust up regulation of these pathways...". As outlined during the first review round, it is unclear whether the mRNA patterns observed in the TTS patients are really caused by continuous up regulation since vaccination, or whether they are only activated when the TTS response started.

Reviewer #2 (Remarks to the Author):

- What are the noteworthy results?

Aid et al. identified markers associated with thrombosis with thrombocytopenia syndrome (TSS) following vaccination with Ad26.COV2.S vaccine (Jansen/J&J), a rare event that occurs in 3 cases per million vaccine recipients that received the Ad26.COV2.S vaccine as initial

vaccination. No TSS cases were reported in the US after the second immunization with the vaccine. The study's authors provide data suggesting that the anti-Ad vector response attenuates the vaccine response and may explain the lack of TSS event after the 2nd immunization.

- Will the work be of significance to the field and related fields? How does it compare to the established literature?

Those are novel and original findings.

- Are there any flaws in the data analysis, interpretation and conclusions?

The number of participants that had TSS in this study is only 2. The authors added a sentence in the discussion listing the sample size as a limitation of the study.

- Does the work support the conclusions and claims, or is additional evidence needed?

The authors addressed most of my concerns with the initial submission.

- Is the methodology sound? Does the work meet the expected standards in your field?

The methodological approaches are sound.

- Is there enough detail provided in the methods for the work to be reproduced?

I still believe that the community would appreciate the custom R code used to perform the analysis and generate the figures being made publicly available to make the data and the results more accessible and reproducible.